# A Survey of Recent Developments in Magnetic Microrobots for Micro-/Nano-Manipulation

**DOI:** 10.3390/mi15040468

**Published:** 2024-03-29

**Authors:** Ruomeng Xu, Qingsong Xu

**Affiliations:** Department of Electromechanical Engineering, Faculty of Science and Technology, University of Macau, Avenida da Universidade, Taipa, Macau, China; yc27952@um.edu.mo

**Keywords:** microrobots, micromotors, microfabrication, magnetic actuation, biomedical application, micro-/nano-manipulation

## Abstract

Magnetically actuated microrobots have become a research hotspot in recent years due to their tiny size, untethered control, and rapid response capability. Moreover, an increasing number of researchers are applying them for micro-/nano-manipulation in the biomedical field. This survey provides a comprehensive overview of the recent developments in magnetic microrobots, focusing on materials, propulsion mechanisms, design strategies, fabrication techniques, and diverse micro-/nano-manipulation applications. The exploration of magnetic materials, biosafety considerations, and propulsion methods serves as a foundation for the diverse designs discussed in this review. The paper delves into the design categories, encompassing helical, surface, ciliary, scaffold, and biohybrid microrobots, with each demonstrating unique capabilities. Furthermore, various fabrication techniques, including direct laser writing, glancing angle deposition, biotemplating synthesis, template-assisted electrochemical deposition, and magnetic self-assembly, are examined owing to their contributions to the realization of magnetic microrobots. The potential impact of magnetic microrobots across multidisciplinary domains is presented through various application areas, such as drug delivery, minimally invasive surgery, cell manipulation, and environmental remediation. This review highlights a comprehensive summary of the current challenges, hurdles to overcome, and future directions in magnetic microrobot research across different fields.

## 1. Introduction

With the development of disciplines such as biomaterials, engineering, and medicine, microrobots are gaining increasing attention [1,2,3,4,5,6]. Due to their distinctive attributes, including compact dimensions, minimal weight, and exceptional flexibility, these robots possess the capacity to undertake tasks beyond the reach of conventional industrial robots [7]. Significant endeavors have been undertaken to develop microrobotic systems endowed with potent transport and delivery capabilities. They can access intricate and confined regions of the human body with minimal invasiveness, enabling micro-/nano-manipulation tasks, such as targeted delivery, precise surgical procedures, and medical examinations [8,9,10,11,12]. This suggests that microrobots have significant potential for applications in the biomedical field.

Various driving technologies have been widely applied to enable the desired movement of microrobots. The actuation strategies, categorized into chemical or external propulsion, involve electric, thermal, chemical, optical, ultrasonic, and magnetic methods [13,14,15,16]. Due to its property of wireless control and the higher degree of freedom for displacement conferred by magnetic forces, magnetic propulsion distinguishes itself among driving methods [17]. Compared to microrobots propelled by alternative methods, magnetic microrobots demonstrate relative safety for biological tissues and exhibit excellent controllability [18,19,20]. Thus, magnetic microrobots have sparked widespread research interest, leading to rapid development in recent years. Simultaneously, the development of smart materials has endowed magnetic microrobots with multifunctionality, various structural designs, and the possibility of employing different magnetic propulsion methods [21,22,23].

This paper comprehensively investigates the recent progress in the field of magnetic microrobots and delves into various aspects such as materials, propulsion mechanisms, design strategies, fabrication techniques, and applications. Additionally, it summarizes the challenges encountered in current research and offers prospects for future development. Figure 1 displays a schematic illustration of the magnetic microrobots discussed in this review and serves as a visual guide to outline the article’s structure. The review initiates with an analysis of magnetic materials. Given the widespread utilization of microrobots in the biomedical field, it is essential to consider the biocompatibility of materials. Following this, there is a discussion of propulsion methods. Different forms of external magnetic fields can induce various motions or functional attributes in magnetic materials. The design section covers multiple types of microrobots: some are categorized based on structure, such as helical, ciliary, and biohybrid microrobots, while others are classified based on functionality, including surface walkers and scaffold microrobots. Microrobots are significantly smaller than traditional robots, and conventional manufacturing methods cannot meet the precision requirements. Subsequently, there is a detailed presentation of fabrication techniques. Finally, the paper concludes by emphasizing the diverse applications of magnetic microrobots for micro-/nano-manipulation, including drug delivery, minimally invasive surgery, cell manipulation, and environmental remediation.

## 2. Materials for Magnetic Microrobots

The choice of microrobot fabrication materials plays a crucial role in their locomotion performance and functional properties, which significantly influences their potential application scenarios [24,25,26]. Therefore, investigating the material characteristics of magnetic microrobots is of significant importance for their development.

Microrobots containing magnetic materials can react to external magnetic fields, allowing for directional guidance and the generation of propulsive forces [27,28,29,30]. Furthermore, to preserve additional characteristics of microrobots, such as structural rigidity, biocompatibility, and biodegradability, it is essential to incorporate non-magnetic materials [31,32,33,34].

### 2.1. Magnetic Materials

For the fabrication of magnetic microrobots, magnetic materials, with their intrinsic property allowing them to be guided by an external magnetic field, have undergone thorough investigation in biomedical applications.

Based on their susceptibility to magnetization in a magnetic field, magnetic materials can be categorized into ferromagnetic [35,36], paramagnetic [37], and antiferromagnetic materials [38]. Ferromagnetic materials are predominantly utilized in magnetic microrobots due to their robust magnetic properties in comparison to paramagnets and antimagnets. Ferromagnetic materials demonstrate significant magnetization in the presence of a magnetic field and can maintain residual magnetism upon the removal of the magnetic field [39]. They exhibit high susceptibility but are inevitably associated with hysteresis effects.

According to the magnitude of coercivity (resistance to demagnetization), ferromagnetic materials can be further classified into hard magnetic (like neodymium—iron—boron, NdFeB) and soft magnetic materials (typically represented by iron and nickel). Both hard and soft magnets exhibit hysteresis behavior, requiring an opposing coercive magnetic field during demagnetization. The relationship between remanent magnetism and coercivity is direct—the higher the remanent magnetism, the higher the coercivity. Consequently, hard magnets possess high coercivity and remanent magnetization, maintaining their magnetization against external fields and making them suitable for permanent magnet applications. In contrast, soft magnetic materials, characterized by low coercivity, can be easily magnetized and demagnetized by external fields [24].

Paramagnetic materials (like ferrite) are also applicable to microrobot fabrication [40]. Unlike ferromagnetic materials, they do not retain magnetization upon removing the external magnetic field. The absence of magnetic hysteresis in paramagnetic materials can reduce potential negative impacts on microorganisms in biological applications. However, similar to antiferromagnetic materials, paramagnetic materials demonstrate reduced sensitivity to magnetic fields compared to their ferromagnetic counterparts. Consequently, their attraction to the same magnetic field is significantly weaker, resulting in limitations in specific applications.

Superparamagnetic materials are characterized by their tiny particle sizes, leading to robust paramagnetism in the presence of an external magnetic field [41]. The coercivity of magnetic materials is significantly influenced by particle size. At sufficiently small sizes, magnetic particles tend to maintain uniform magnetization and resist the formation of magnetic domains, displaying single-domain characteristics with high remanence. Consequently, as the sizes of ferro- or ferrimagnetic particles decrease below a critical size, their remanence diminishes without an external magnetic field (due to decreasing coercivity). However, in the presence of a magnetic field, superparamagnetic particles retain relatively high magnetic susceptibility [42,43].

Superparamagnetic materials are commonly utilized in the fabrication of ferrofluidic robots, which are alternatively referred to as active drops owing to their liquid properties and capacity to maneuver under the influence of an external magnetic field [44,45]. Ferrofluidic robots offer superior capabilities over elastomeric-based soft robots for navigating narrow and confined spaces, thereby minimizing damage to surrounding biological tissues during biomedical procedures and achieving the least invasive approach [46,47].

If preparing droplets at the micrometer and nanometer scales, special methods, such as microfluidic synthesis, are required. Microfluidic synthesis is a method that utilizes microchannels and microfluidic technology to synthesize microparticles, microstructures, or microdevices [48]. On a microfluidic platform, precise control over reaction conditions and substance transport is achieved by manipulating fluid flow, mixing, and reaction processes within microchannels. It allows for precise control and fabrication of microstructures. This method is commonly employed for the preparation of microdevices with specific functionalities and properties, such as microcapsules, nanoparticles, microfluidic chips, etc., and finds wide applications in the biomedical field, chemical engineering, materials science, and other domains [49].

Due to the absence of the aforementioned magnetic hysteresis effect, superparamagnetic materials do not exhibit remanence without an external magnetic field. This significantly reduces the potential side effects of magnetic materials within biological organisms, making them commonly applied in the field of biomedical research.

The magnetic moments of antiferromagnetic materials align along specific directions under an applied magnetic field, but the directions of adjacent magnetic moments are opposite. This arrangement results in the material’s magnetization being zero, even under an applied magnetic field, and it does not exhibit significant magnetism. Due to this characteristic, antiferromagnetic materials are not used in the fabrication of magnetic robots. However, study on antiferromagnetic materials is also significant in the fields of materials science and magnetic materials research, as it holds value for understanding the magnetic behavior of materials and developing new types of magnetic materials.

Most small robots consist of ferromagnetic and superparamagnetic materials, as indicated in Table 1. The widespread use of ferromagnetic compounds, including Ni, Fe, and NdFeB, is credited to their elevated saturation magnetization and their ability to propel and control microrobots under low magnetic field intensities [50,51]. Higher saturation magnetization leads to increased magnetic force in the magnetic field, resulting in enhanced maneuverability.

### 2.2. Biosafety Materials

Microrobots are extensively employed in biomedical applications due to their small size [41,73,74,75]. The selection of materials should carefully consider both biocompatibility and biodegradability. Otherwise, microrobotic systems may trigger immune responses or inflammation in the body [9,76,77,78]. Before contemplating the actual deployment of microrobots in practical scenarios, it is imperative to meticulously investigate and rectify the biocompatibility, potential biodegradability, and interaction dynamics with the biological elements inherent in such microrobotic systems. Subsequent sections will delineate the endeavors undertaken by the microrobotics community in pursuit of these objectives.

#### 2.2.1. Biocompatible Materials

Biocompatibility is one of the foremost requirements in the design of microrobots for biomedical applications [79]. It not only diminishes the need for microrobot recycling but also prevents inadvertent residues of microrobots [80]. So far, there are two methods for establishing biocompatibility. One approach combines the use of biocompatible materials or the creation of protective coatings [81,82,83,84], while the alternative approach involves the development of biohybrid microstructures. This includes biohybrid microrobots, which consist of artificial constructs affixed to biological cells [85,86], or artificial microrobots enveloped in living cell membranes [87].

Coatings with protective properties—including those derived from biodegradable copolymeric compounds such as PLGA (poly(lactic-co-glycolic acid)) and copolymers containing methacrylic acid, along with natural polymers like chitosan—have exhibited positive biocompatibility in diverse cargo delivery approaches in the gastrointestinal tract (GI) [88,89]. The initial demonstration of in vivo therapy featured microrobots with a magnesium core coated with a layer of antibiotic-loaded PLGA and an outer chitosan covering, ensuring microrobot adherence to the stomach wall through electrostatic interactions. This microrobot design based on PLGA-chitosan composition led to prolonged retention on the stomach wall, thereby augmenting the effectiveness of therapeutic interventions [10].

Furthermore, researchers have explored inorganic coverings, including Ti and Fe. Nelson’s group utilized a burr-like porous spherical structure coated with Ti material to carry and deliver to targeted cells in vivo, ensuring biocompatibility [90]. However, despite ongoing investigations, exposure risks persist during the fabrication process, and the coating layer remains incapable of completely enveloping microrobots.

Derived from polyethylene glycol (PEG), PEGDA (poly(ethylene glycol) diacrylate) is hydrophilic and elastic and boasts favorable photopolymerization characteristics that facilitate the facile fabrication of microrobots with defined configurations for applications in regenerative therapy and tissue regeneration [91,92]. The mechanical properties of the resulting products are modifiable, with the elastic modulus increasing in tandem with the concentration of PEGDA [93]. Liu et al. recently demonstrated the preparation of soft microhelixes with adjustable mechanical characteristics by incorporating PEGDA with alginic acid calcium salt [94]. By manipulating the PEGDA concentration within the initial formulation, one can effortlessly refine the elastic moduli of the pliable microrobots to span from tens of kPa to a few MPa. In achieving the construction of autonomously moving microrobots featuring filamentous hydrogel tails derived from PEGDA, Srivastava et al. utilized live-time, on-site polymerization [95]. The polymeric structure of PEGDA, resembling a thread, possesses low mechanical strength, making it easily bendable and deformable. This characteristic enables the remote capture of cells and microparticles. Furthermore, PDMS and Ecoflex silicone rubbers have been demonstrated to exhibit biocompatibility [96,97,98].

Developing biohybrid microstructures offers an alternative way to establish biocompatibility. The amalgamation of microrobots incorporating components from native cells enhances the overall compatibility of the microscale robotic system with biological organisms, thereby diminishing immune responses. Essentially, the collaborative interaction between the effective mobility of artificial microrobots and the inherent biological characteristics and functions of cellular elements has resulted in the development of microrobots that mimic cells and exhibit inherent biological functionality. This includes capabilities such as toxin and pathogen removal and holds significant promise for therapeutic applications. A recent design involved testing a new magnetic nanopropeller, whereby Fe and Pt were co-deposited onto silicon dioxide, for the delivery of plasmids into cells. The results demonstrated non-toxicity to cells and outstanding biocompatibility [56].

#### 2.2.2. Biodegradable Materials

While biocompatible materials may not completely avoid all safety issues in biomedical applications, the potential presence of residual microrobot components within the body could lead to adverse effects.

Manufactured through an economical production method employing a biological template derived from the cyanobacterium Spirulina, a porous microrobot with a hollow structure is characterized by an outer shell composed of superparamagnetic Fe_3_O_4_ and an inner cavity with a helical shape. This microrobot, determined to be compatible with living cells and capable of natural degradation, can be disassembled into smaller fragments under ultrasonic stimulation [99].

While opting for biocompatible or biodegradable materials as the primary constituents for microrobots is a prudent choice for potential clinical applications, the most widely employed strategy still involves using a photosensitive polymer serving as the structural foundation and covered with a coating of materials compatible with living organisms. This approach represents a compromise by considering the manufacturing method’s complexity and the materials’ inherent mechanical robustness. For instance, a magnetic microrobot fabricated using SU-8 (a negative photoresist) with a tailored microstructure created through 3D laser lithography is then covered with a layer of Ni for magnetic manipulation and an extra layer of Ti or Pt to enhance biocompatibility [68].

Since the primary constituents of these microrobots are not subject to natural degradation, there is a risk of accumulation in the body, potentially causing severe inflammation in the absence of an appropriate recycling mechanism. Nevertheless, polymers can be easily fashioned into intricately designed microstructures using 3D laser printing and include both the required hardness and precision of the structure as well as biocompatibility. Consequently, one crucial direction is to develop a 3D manufacturing method based on hydrogel materials to advance the clinical application of cell-transport microrobots.

### 2.3. Current Challenges and Prospects

To obtain materials that are better suited for microscale robot performance, expertise in multidisciplinary fields such as materials science and electromagnetics is required. While existing magnetic materials generally meet current application demands, achieving precise control for medical applications remains a significant challenge. The most widely used ferromagnetic materials exhibit residual magnetism even in the absence of a magnetic field, which is accompanied by hysteresis effects. In applications requiring precise control of magnetic material magnetization, hysteresis effects may result in insufficiently prompt changes in magnetic field intensity, thereby affecting system response speed and accuracy [100,101,102]. Hysteresis effects pose obstacles to achieving precise control with ferromagnetic materials, inevitably leading to energy loss and potentially affecting material lifespan. Paramagnetic materials, although devoid of hysteresis effects, exhibit low sensitivity to magnetic fields, making them less responsive to subtle magnetic field changes. Moreover, they require relatively large magnetic fields for driving and are subject to certain limitations in applications. In the future, exploring magnetic materials that combine the advantages of both materials could facilitate precise control.

The biosafety of microrobots represents a higher pursuit, with current obstacles primarily stemming from fabrication methods. For instance, as mentioned earlier, coating methods face challenges in achieving complete encapsulation without compromising precise magnetic manipulation in a magnetic field.

## 3. Propulsion of Magnetic Microrobots

When subjected to an external magnetic field, magnetic microrobots undergo both a magnetic force, denoted as $F_{m}$, and a magnetic torque, $T_{m}$ [103]. The expression for the magnetic force acting on the microrobot is:(1)$$F_{m}=V\left(M\cdot ▽\right)B$$
where *V* and M represent the volume and magnetization of the microrobot’s magnetic material, respectively. B signifies the magnetic flux density of the external magnetic field.

Simultaneously, the magnetic torque $T_{m}$ can be mathematically expressed as:(2)$$T_{m}=VM\times B$$

As the magnetic microrobot is exposed to the magnetic field, it becomes magnetized, generating magnetization in the process. For a linear isotropic media, the magnetization is influenced by the external field and follows the relationship between M and H:(3)$$M=\chi_{a}H$$
where $\chi_{a}$ and H denote the susceptibility tensor of the material and the magnetic field strength, respectively.

In the case of a homogeneous magnetic field (where the gradient ▽ vanishes), the magnetic robot does not encounter a gradient force and typically moves parallel to the field. However, $T_{m}$ possesses the capability to force the magnetic microrobot to orient its dipole moment in alignment with the external magnetic field through rotational motion, especially when the microrobot and the magnetic field are not oriented in the same direction [104]. Therefore, magnetic fields employed for propelling microrobots should be time-invariant (such as rotating and oscillating magnetic fields) or inhomogeneous (e.g., a gradient magnetic field).

### 3.1. Rotating Magnetic Field

Electromagnetic coils are common devices for generating a rotating magnetic field. In comparison to microrobots activated by magnetic field gradients and oscillating magnetic fields, those propelled by rotating magnetic fields exhibit superior maneuverability and precise locomotion [105,106]. In ideal conditions, they can efficiently operate even in magnetic fields with zero field strength.

Generally, Maxwell coils and Helmholtz coils are commonly used devices for generating magnetic fields; both consist of multiple coil sets. Although they do not directly produce rotating magnetic fields, the desired effect of a rotating magnetic field can be indirectly simulated by appropriately adjusting the direction and intensity of the electric current. They are commonly utilized for driving helical robots, whereby the actuation is accomplished by initiating rolling, corkscrew, and spin-top movements [107,108,109]. By engaging in rotation around the helical axis, these microrobots progress in the direction perpendicular to the rotation plane of the microrobot. Much like their helical counterparts, magnetized spherical microrobots can also effortlessly respond to rotating magnetic fields as well [110]. In 1996, Honda et al. introduced a propulsion method utilizing a rotating magnetic field that involved a square-shaped magnet affixed to a coiled copper conductor [111]. Demonstrating successful propelling in silicone oil with elevated viscosity at a low Reynolds number, the magnetically actuated helical wire showcased its suitability for microscale propulsion. The experimental findings of their study indicated the effectiveness of rotating-magnetic-field-based propulsion methods in microscale environments. Derived from the bacterial flagella movement, Zhang et al. introduced a simulated helical flagellum that is propelled by rotational magnetic fields in a microscale fluidic environment [112]. Consisting of a helical appendage and a slim magnetic head formed via the self-scrolling of a metallic double layer, the artificial flagellum mimicked the motion of natural counterparts. It has been reported that rotating magnetic fields not only manipulate magnetic particle aggregates but also control various other intriguing structures [72,113,114].

### 3.2. Gradient Magnetic Field

The methods for achieving a gradient magnetic field include employing permanent magnets with position control [115], utilizing electromagnetic devices with the ability to adjust their location by manipulating applied electrical currents [116,117,118], and applying stationary electromagnets with controlled electrical signals [119,120]. Using permanent magnets to achieve gradient magnetic fields is relatively simpler compared to the other two methods. By adjusting the position and orientation of permanent magnets, different magnetic field gradients can be produced in space. This can be achieved through mechanical approaches or magnetic positioning systems. If electromagnetic devices or static electromagnetic systems are used, precise position control systems and current regulation devices are required to ensure that the magnetic field intensity and direction can be adjusted and controlled as needed.

In contrast to microrobots propelled by a rotating or an alternating magnetic field, those activated by magnetic field gradients have fewer constraints on their mechanical structures and can travel in alignment with the field gradient. Magnetic-field-gradient-driven microrobots often adopt spherical and cylindrical structures, as these configurations experience minimal surface frictional forces.

### 3.3. Oscillating Magnetic Field

The methods for generating an oscillating magnetic field include driving electromagnets or coils with alternating current power sources as well as utilizing specific circuits and control systems to induce periodic variations in the magnetic field [110,121].

The generation of non-reciprocal motion in magnetic materials is attributed to oscillating magnetic fields and is identified as magnetic vectors varying over time and moving vertically within a plane [122,123]. This non-reciprocal motion is a crucial feature essential for propulsion methods employing time-varying magnetic torque. In contrast to microrobots propelled by a rotating magnetic field or magnetic field gradients, those propelled by oscillating magnetic fields employ asymmetrical shape deformation to overcome Purcell’s renowned “scallop theorem” [99]. The prevalent design for microrobots driven by oscillating magnetic fields is a multijoint structure, drawing inspiration from the swimming mechanisms observed in fish or bacterial flagella [39,124].

### 3.4. Current Challenges and Prospects

Due to the increasing popularity of magnetic propulsion research, an expanding array of magnetic propulsion systems has been developed, with many commercially deployed in medical and other fields. Currently, there are two primary types of magnetic propulsion systems: electromagnetic systems and permanent magnet systems. Electromagnetic systems encompass multi-axis Helmholtz coils [125], OctoMag [126], and Maxwell [127]. Permanent magnet systems include single-magnet systems and multi-magnet systems such as the commercially available Niobe system [128].

Although a considerable number of magnetic propulsion systems have been commercialized, several issues remain unresolved. First, the penetrability of these existing magnetic fields is a significant concern. Current magnetic fields used for precise manipulation often require substantial magnetic strength, resulting in high energy consumption and, consequently, elevated costs. Additionally, intense magnetic fields may pose side effects on the human body.

Second, the practicality of current magnetic fields requires improvement. Many commercially available magnetic propulsion systems have limited applications. For instance, the system proposed by Ankon Technologies is currently only employed for propelling magnetic endoscopes [129], while the Niobe system is utilized for guiding catheters in cardiovascular disease treatment.

Microrobots hold vast potential in the fields of cluster control and three-dimensional control. Primarily, cluster control enhances the efficiency and agility of microrobots in task execution [130]. Through collaborative efforts, they can jointly accomplish intricate tasks such as environmental monitoring and search and rescue operations, thus augmenting work efficiency. Additionally, advancements in three-dimensional control technologies empower microrobots to freely navigate in complex three-dimensional spaces, enabling exploration of uncharted territories and unlocking new application potentials in domains like healthcare, construction, and environmental monitoring [131].

Nevertheless, microrobots encounter numerous challenges in both cluster control and three-dimensional control. Firstly, cluster control necessitates addressing issues such as communication, localization, and path planning. The diminutive size and limited computational capabilities of microrobots render cluster cooperation increasingly intricate. Secondly, precise control of microrobots in three-dimensional space requires consideration of factors like gravity and air resistance. Moreover, microrobots are susceptible to environmental disturbances during movement, further complicating control efforts. These challenges, in turn, constrain their practical utility.

In summary, future research endeavors will primarily focus on achieving strong penetrability and multifunctional practicality.

## 4. Design of Magnetic Microrobots

Due to the constraints imposed by fabrication methods and the distinct physical phenomena dictated by low-Reynolds-number hydrodynamics at the microscale, reducing macroscopic actuation mechanisms for generating microscale motion is often impractical [132].

### 4.1. Helical Microrobots

In a low-Reynolds-number environment, microrobots achieve effective propulsion by producing anet unidirectional movement through asymmetric motion. Inspired by the bacterial flagellum, a design approach that incorporates magnetic helical structures on microrobots enables propulsion by applying magnetic torques within the body under a weak rotating magnetic field [133]. Typically, a helical microrobot consists of either a magnetic helical tail or one affixed to a magnetic head. This design facilitates a corkscrew-like motion, translating rotational motion around the helical axis into nonreciprocal translational movement [134,135]. Alteration of the external magnetic field’s counterclockwise or clockwise direction makes it easy for helical robots to attain forward or reverse movement [13].

In 2009, Abbott et al. conducted a comparative analysis of three promising methods for microrobot swimming, with the magnetic propulsion of helical structures achieving swimming motion for the first time [110]. Subsequently, helical microrobots have found increasing application in the biomedical field. Proposing a degradable superparamagnetic polymer composite, Peters et al. offered safe degradation of the device in vivo, followed by the non-invasive excretion of degradation products, presenting an alternative to manual or surgical device recovery [83]. The results of in vitro biodegradation experiments are presented in Figure 2a. Subsequently, Ceylan et al. designed a water-based, enzyme-degradable dual-helical microswimmer propelled and controlled by magnetic forces [41]. Under rotational magnetic fields, this microrobot can load cargo volumetrically and exhibit swimming capabilities. The 3D microswimmer was optimized for performance and was 3D-printed using two-photon polymerization. The printing process utilized a gelatin methacryloyl-based magnetic suspension and biofunctionalized magnetic iron nanoparticles with superparamagnetic properties. Figure 2b presents the dual-helical structure of the microrobot. Nowadays, an increasing number of helical microrobots are addressing specific challenges in medical applications. For instance, a microrobotic hierarchical superstructure was proposed in 2023 by Landers et al. [136]. Comprising magnetic helical micromachines intricately connected to a thermally responsive transient magnetic polymer framework, these formations are designed to intelligently navigate and transport small magnetic helical micromachines to intricate small vessels and capillaries, thereby facilitating access to challenging anatomical sites within the human body. As shown in Figure 2c, the microrobot can traverse various shapes of microchannels under the influence of an external magnetic field. Liu et al. successfully proposed a micromotor through microfluidic chips, as illustrated in Figure 2d [137]. Capable of loading neuron stem cells and precisely delivering them to targeted areas under external magnetic fields, the micromotor also contributes to restoring the interconnection of neurons. This research introduces a novel approach for reconstructing neural networks in cases of neuron injuries.

### 4.2. Surface Microrobots

Surface walking has emerged as a distinctive propulsion method in recent microrobots applications. An illustrative example is the surface-walking Janus microdimer, a microsurface walker activated by magnets and consisting of two magnetically connected Janus spheres made of Ni/Si$O_{2}$ [138,139]. When subjected to a flat, undulating magnetic field near a surface, these spheres exhibit an asymmetric rolling motion over each other, resulting in a net displacement. This innovative approach introduces new possibilities in microrobot propulsion, particularly for navigating confined spaces and intricate geometries. Due to its robust navigational capabilities and efficiency in speed modulation, developers envision diverse applications for this device ranging from nanomanipulation to precision medicine treatments.

### 4.3. Ciliary Microrobots

Magnetic microrobots employing propulsion mechanisms based on traveling waves and metachronal waves draw inspiration from ciliate microscopic organisms in nature, such as *Paramecia*. This robotic type disrupts temporal symmetry, achieving effective movement through phase discrepancies in the beating motion of neighboring cilia [140].

In 2007, Evans et al. developed a method to create arrays of high-aspect-ratio cantilevered micro- and nanorods using a PDMS-ferrofluid composite material, as shown in Figure 3a [141]. Belardi et al. introduced a novel process for producing large arrays of magnetically driven artificial cilia by utilizing a two-color lithography-based approach in 2011 [142]. The exploration of methods for fabricating cilia has contributed to the development of ciliary microrobots. Furthermore, multi-joint microstructures can generate propagating wave or metachronal undulation propulsion. In 2018, Li et al. designed a magnetic artificial multi-legged millirobot [65]. Showcasing exceptional adaptability in various challenging environments, this robot design demonstrates remarkable features such as rapid mobility, high carrying capacity, and outstanding obstacle-crossing ability. The gait of the microrobot is illustrated in Figure 3b.

Recently, Wei et al. designed a ciliate microrobot using PDMS and NdFeB that mimics an *inchworm*’s gait and can move on wet surfaces and inclines. The structure of the robot is illustrated in Figure 3c [143]. Using soft miniature devices, Dong et al. investigated the quantitative relationship between metachronal coordination and consequent fluid dynamics [144]. Additionally, they designed fluidic soft devices inspired by cilia with distinctive capabilities for maneuvering and blending thick synthetic and biological fluids in low Reynolds numbers, as depicted in Figure 3d. Xu et al. proposed a swarming method that utilizes physical interactions among magnetic microparticles to arrange them into cilia structures, as shown in Figure 3e [145]. Ciliate microrobots have additional applications. In 2023, Feng et al. incorporated cilia structures into sensor modules, enabling the detection of signals related to joint movements and the ability to sense pressure applied to the skin [146]. The cilia structure, fabricated using carbonyl iron particles, aids with navigation or sensing within the targeted environment. The structure is depicted in Figure 3f.

### 4.4. Scaffold Microrobots

Drawing inspiration from the porous extracellular matrix, scientists have created 3D microscaffolds with linked pores that act as platforms for cell and tissue cultivation [147,148]. The microscaffolds, which form the basis of magnetically actuated microrobots, offer significant capacity for loading and storing. Playing a vital role in enhancing targeted drug delivery, manipulating cells, and performing minimally invasive surgery, they contribute significantly to efficacy improvement [149,150,151]. In 2017, Go et al. introduced magnetically actuated microscaffolds designed to carry mesenchymal stem cells (MSCs) for repairing joint cartilage. Represented as microspheres with interconnected micro holes, these 3D microscaffolds provide sufficient room to facilitate cell adhesion, provide support, and enable transportation. Besides spherical microscaffold structures, researchers have designed microscaffolds with various shapes by employing state-of-the-art 3D-printing methods. For instance, Jeon et al. created many microscaffold structures characterized by complex shapes and consistent pore sizes. These structures involve cylindrical, helical, and rectangular objects [67,73].

### 4.5. Biohybrid Microrobots

Over millions of years, biological systems have evolved to operate optimally at the microscale and showcase remarkable movement and functionality. In the past decade, the development of magnetic microrobots has gained attention, with biohybrid systems emerging as attractive approaches. Currently, there is significant interest in biohybrid miniaturized motors, which are characterized by outstanding biocompatibility and minimal toxicity. Consequently, biohybrids present promising alternatives. Comprising two essential elements, biohybrid robots consist of living biological entities or tiny organisms that exhibit optimal biocompatibility and deformability as the first component. The second component involves artificial microstructures or microparticles, which serve as carriers to support these cells [152,153]. These micromotors typically combine synthetic microstructures with non-mobile cells or mobile cells [2,154].

#### 4.5.1. Microrobots Based on Nonmobile Cells

Certain plant cells, like pollen and spores, can be employed to fabricate biohybrid robots. Generally, pollen and spores exhibit excellent biocompatibility features and structural uniformity. Additionally, specific plant cells, such as those with unique architectures like pollen cavities, have the potential to serve as carriers for increased cargo capacity [155,156,157].

Utilizing a vacuum loading technique, Sun et al. successfully embedded drugs into the hollow section of pine pollen grains [72]. The experiments showcased the remarkable drug-encapsulation capability of the pollen-based magnetic microrobots and highlighted their effective containment capability and precision in drug discharge. Driven by programmatically controllable magnetic fields and containing encapsulated magnetic Fe_3_O_4_, these micromotors exhibited three different movement modes: rolling, crawling, and rotating. Moreover, under the influence of an external magnetic field, individual pollen-based micromotors could collectively form dynamic phenomena. The dynamic phenomena refer to the controlled release of therapeutic cargo achieved through the manipulation of magnetic nanoparticle aggregation under a powerful magnetic field. This phenomenon is crucial as it enables precise and controlled drug delivery, which is essential for various biomedical applications.

Additionally, spores offer another avenue for the synthesis of magnetically actuated biohybrid microrobots [158,159]. In a study conducted by Zhang et al., magnetic nanoparticles were directly deposited, followed by the encapsulation of functionalized carbon nanodots on porous natural spores, resulting in the synthesis of magnetic spore-based microrobots [159], as shown in Figure 4a.

#### 4.5.2. Microrobots Based on Mobile Cells

In nature, cell structures endowed with mobility not only serve as inspirational sources for the intricate designs of microrobots but also offer numerous functional advantages when integrated into microrobots. These advantages encompass attributes such as biodegradability, ease of absorption, and spontaneous fluorescence. Expanding the biological applications of magnetic robots is achievable by incorporating these intriguing functionalities into their fabrication. The common biohybrid microrobots are depicted in Figure 4.

One method for creating hybrid microrobots involves the combination of active locomotive cells, particularly those possessing flagella, with sperm and bacteria being commonly employed for this purpose [85,165,166,167]. In this approach, the motile cell can either attach to the surface of a synthetic particle or another cell or become ensnared within a specialized microstructure. Under a rotating magnetic field, a self-assembled magnetic nanorobot created by Ali et al. utilizes bacterial flagella attached to a superparamagnetic particle for actuation and steering [160], as depicted in Figure 4b. Due to the high binding affinity of doxorubicin hydrochloride (DOX-HCl), i.e., a hydrophilic anticancer drug, to DNA (nucleus), sperm were found to exhibit significant uptake of DOX-HCl in a study [166]. With the goal of designing a drug-loading microrobot aimed at targeted cancer treatment, Magdanz et al. employed bovine sperm cells as a template to fabricate soft magnetic microrobots through a straightforward electrostatic-based method (Figure 4d) [162].

Another approach involves utilizing live immune cells to engulf entire passive magnetic functional particles, leveraging the phagocytosis processes of immune cells [168]. When macrophages completely engulf the magnetic double-helix micromotor, the microrobot exhibits rolling motion driven by magnetism along a predetermined trajectory, propelling the magnetic spiral components. These robots showcase the capability of continuous swimming even when facing obstacles from cells. Without an external magnetic field, the immunobot autonomously moves by creeping, driven by the self-propulsion locomotion of macrophages in vivo [169].

In addition, Yan et al. discovered that *Spirulina platensis*, a helical microalgae subspecies, exhibits intrinsic fluorescence, selective cytotoxicity against cancer cells, and natural degradability [170]. By employing a dip-coating method to attach Fe_3_O_4_ nanoparticles to the surface of *Spirulina platensis*, they produced magnetic microrobots collectively known as magnetized *Spirulina*.

### 4.6. Current Challenges and Prospects

Researchers have designed various types and sizes of microrobots to accommodate different application scenarios. Considering applications in the medical field, there are still some barriers to overcome. Produced through specific fabrication methods (discussed later), current magnetic microrobots can achieve sizes ranging from nanometers to millimeters. However, a significant challenge arises: how can microrobots of such small dimensions resist fluid flow? Currently, there are two main approaches to address this issue.

Firstly, increasing the external magnetic field is one option, but it comes with high costs and difficulty with providing sufficient driving force due to the small size of the robots. Second, adapting to different flow velocities by designing robots with various sizes presents a compromised solution. However, increasing the size may lead to the issues such as vascular blockages. In the future, hopefully there will be specific structural designs to mitigate these problems.

As microrobots continue to gain widespread adoption in biomedical settings, the importance of biocompatibility is expected to grow significantly as a focal point of future research. Biohybrid microrobots, benefiting from biological components, exhibit enhanced biocompatibility. It can be boldly speculated that there will be an increasing number of biohybrid microrobots tailored to diverse intracorporeal environments in the future. Furthermore, these biohybrid microrobots hold potential to address the challenges related to crossing the blood–brain barrier, which is traditionally considered a formidable obstacle in biomedical interventions.

## 5. Fabrication of Magnetic Microrobots

After the discussion of various magnetic microrobots, it is also crucial to understand how they achieve their functions through fabrication. Due to the vast differences in structure and materials between magnetic microrobots and industrial robots, the development of suitable fabrication methods for microrobots is urgently needed. Currently, common preparation methods for magnetic microrobots include direct laser writing (DLW), glancing angle deposition (GLAD), biotemplating synthesis (BTS), template-assisted electrochemical deposition (TAED), and magnetic self-assembly (MSA).

### 5.1. Direct Laser Writing (DLW)

Direct laser writing (DLW) eliminates the need for a mask plate for employing two-photon absorption to induce changes in the solubility of the resist and developer, distinguishing it from conventional lithography techniques [66,171,172]. Utilizing two-photon polymerization (TPP) nanolithography, DLW enables the production of intricate 3D microscopic-scale structures with exceptionally high spatial resolution.

In a study by Wang et al., the locomotion of spiral microrobots with different surface coatings and exhibiting varying degrees of wetting on their surfaces was investigated [173]. The process involved laser-writing of the photoresist on a glass substrate to create helix structures. Subsequently, the polymer microhelices were coated with nickel for magnetic propulsion, followed by a gold coating. This was achieved through physical evaporation to enhance adhesion during subsequent surface modification. Dong et al. utilized 3D laser lithography based on TPP and water dispersion to construct helical microrobots using hydrogel materials [174]. Subsequently, these printed spiral microrobots are coated with magnetic particles on their surfaces, forming motion components driven by an externally rotating magnetic field. In a study by Ceylan et al., the TPP process was employed to modify a superparamagnetic ferrite matrix, significantly increasing the volume-to-surface area ratio [41]. Additionally, Sun et al. designed and fabricated novel burr-like porous spherical structures using the TPP process; the resulting structures showcased superior cellular delivery efficiency in both physiological and external environments [68].

### 5.2. Glancing Angle Deposition (GLAD)

As an extension of conventional physical vapor deposition methods, glancing angle deposition (GLAD) involves the synchronous manipulation of the substrate with incident vapor deposition, deviating from the typical practice of deposition on a fixed substrate [81].

In the literature, Walker et al. established a monolayer using 500 nm silicon dioxide on a silicon chip as seed nuclei for nucleation points. Subsequently, nickel was deposited onto the silicon particles at a specific angle for magnetostriction purposes [82]. Simultaneously, Venkataramanababu et al. created a 2D array of photoresist pillars through laser interference lithography. Following the deposition of the seed layer, it enhanced the shape customization capability of spiral robots [175].

### 5.3. Biotemplating Synthesis (BTS)

As mentioned earlier, recent studies demonstrate that biohybrid microrobots, formed through the combination of suitable microorganisms with artificial micromaterials, maintain their outstanding biological characteristics (including biocompatibility and biodegradability) during migration in a low-Reynolds-number fluid. Consequently, the biotemplating synthesis (BTS) approach enables microscale robots to possess healing capacities for various medical uses in a safe and biocompatible fashion. The state-of-the-art literature showcases similar endeavors in microrobots, such as combining with pollen, bacteria, and cells [176,177]. Notably, the natural microalgae *Spirulina* (Sp.) stands out due to its remarkable 3D helical microarchitecture and frequently serves as a biotemplate for constructing biohybrid helical microrobots for activities such as transporting goods or purifying water [170,178,179].

The magnetic microrobots based on microalgae recently developed by Wang et al. have significantly enhanced swimming speed under a rotating magnetic field [163]. Another potential candidate for constructing cell-based microrobots is red blood cells (RBCs), which lack nuclei, providing a more spacious environment, and are equipped with hemoglobin for oxygen transport [164]. Emphasizing the double-concave structure of RBCs, Gao et al. highlighted their ability to navigate more accurately with the assistance of an external magnetic field and accommodate a greater amount of photosensitizers [71].

### 5.4. Template-Assisted Electrochemical Deposition (TAED)

Template-assisted electrochemical deposition (TAED) represents a straightforward yet efficient approach to form a firmly adhering layer of specified materials onto a pre-made conductive substrate. Due to its low cost, rapidity, and ease of operation, this deposition technique is considered suitable for mass production of microstructures. With the assistance of well-designed templates, diverse microstructures, such as helical [180], tubular [181], and rod-like [182], have been successfully deposited. Presenting a tri-segmented strategy, Jordi Sort et al. embedded CoPt/Cu/Ni into polycarbonate nanorods through electrochemical deposition [183]. By incorporating semihard magnetic materials to coat nanowires using an anodic aluminum oxide (AAO)-template demolding and premagnetization process, Jang et al. enabled the proposed nanorobots to execute tumbling, procession, and rolling at various rotating frequencies under a rotating magnetic field [184].

### 5.5. Magnetic Self-Assembly (MSA)

In addition to the operation of magnetically powered microscopic robots as individual units, there are recent explorations into active magnetic particles displaying collective or swarm behaviors for on-demand in vivo applications [185]. Magnetic self-assembly (MSA), recognized for its versatile multifunctionality achieved through controllable structure reconfiguration, is considered an effective alternative for fabricating swarm robots. Leveraging a 2D flat-plane rotating magnetic field, Tasci et al. engineered superparamagnetic beads capable of self-assembling into size-controlled microwheels [186]. In addition to widely employed colloidal particles, Wang and colleagues explored magnetic droplets made from a water suspension containing benzyl–ether and carbonyl iron microparticles [187]. These ferromagnetic particles settled at the bottom of the droplet due to gravity and quickly formed a chain when subjected to a processing field.

### 5.6. Current Challenges and Prospects

Current fabrication methods have largely met the basic requirements of magnetic microrobots. However, as the fabrication methods are intricately linked with structural design and material selection of robots, future exploration may lead to the discovery of alternative methods to meet the performance demands of novel microrobots. Additionally, there is a prospective direction for future development in the realm of more convenient and automated fabrication processes. While a few self-assembling magnetic microrobots have been reported, the realization of in vivo self-assembly structures has yet to be achieved. The development of in situ manufacturing techniques could effectively mitigate various issues that may arise during the transportation of microrobots. Hence, it represents a promising avenue for future development.

## 6. Applications of Magnetic Microrobots

In 1959, Richard Feynman first introduced the concept of miniaturized machines during a lecture [188]. Subsequently, in the 1966 film “Fantastic Voyage”, a man ingested a doctor reduced to microscopic size, and the doctor repaired damage in the person’s brain. In recent years, scenes like this have transcended the realm of science fiction. With the advancement of microrobots, their applications have become widespread, particularly in the medical field with micro-/nano-manipulation [189,190,191], as shown in Figure 5.

Discussed in the preceding sections are the design mechanisms and actuation methods of microrobots, which serve as the foundation for their application in various scenarios. The following section will showcase the various applications of microrobots.

### 6.1. Drug Delivery

Transporting therapeutic cargoes with precision and efficiency to specific sites in the body, especially the complicated and confined environments, poses a challenge for passive drug delivery systems. Compared to traditional passive transport, microrobots can achieve targeted drug delivery, enhancing the percentage of drug reaching the diseased site while simultaneously reducing the side effects on normal tissues. The delivered cargoes can be cells, drugs, or other nanoparticles [99,166,194]. Magnetic microrobots have numerous advantages that make them effective carriers for targeted drug delivery. These advantages include the previously mentioned untethered operation, precision, minimal invasiveness, and the recyclability of the robots to minimize residual effects on the human body as much as possible [195,196,197].

Developed for encapsulating or transporting cargoes, magnetic microrobots encompass a variety of micromaterials. These include various organic or inorganic artificial and biogenic materials, such as hydrogel-based helical microswimmers [41], Janus microparticles [62], bacteria [86,176,198], sperm cells [199], and microalgae [200,201].

In 2008, Burdick et al. designed magnetic nanomotors containing nickel that were capable of magnetically controlled cargo manipulation. These nanomotors can achieve loading, dragging, and release of cargoes along their path under the influence of a magnetic field [202]. In 2015, Denzer et al. conjugated antibodies with drug molecules on the surface of magnetic Janus microspheres. This Janus microsphere can precisely target and deliver drugs to cancer cells [203]. Subsequently, Ceylan et al. designed a double-helical structure microrobot capable of cargo loading under the manipulation of a rotating magnetic field [41]. Recently, Akolpoglu et al. reported a miniature robot based on *Escherichia coli* (*E. coli*) that can target and release drugs through a multi-stimulus response [204].

### 6.2. Minimally Invasive Surgery

Miniature robots can be designed as surgical tools to directly penetrate or retrieve cellular tissues [205]. These freely moving minimally invasive systems can access body tissues beyond the reach of blades and catheters. Moreover, they are expected to reduce the risk of infections and to shorten recovery time [206]. In fact, miniature robots are poised to serve as a complement to existing techniques by enhancing the precision and control capabilities of current surgical robotic tools, thereby augmenting the surgical skills of surgeons. By enhancing the penetrative capability of magnetic fields, miniaturized machines can navigate deep tissues or even capillaries remotely, rendering them a promising approach for minimally invasive surgery [207,208].

Magnetic microrobots equipped with pointed ends or capable of executing corkscrew-like movements under a rotating magnetic field can effectively perform drilling procedures. Leveraging this drilling capability holds tremendous potential for achieving untethered microsurgeries with high precision, particularly for penetrating tissues [209]. Additionally, surface walkers exhibit the capacity to open cell membranes, further expanding the applicability of these microscale systems [210,211].

Wu et al. reported the first microscale propeller capable of penetrating the vitreous humor and reaching the retina. Propelled by an external magnetic field, the spiral propeller with the surface coated by perfluorocarbon reached the retina within 30 min [81]. In 2022, Vyskocil et al. proposed a type of Au/Ag/Ni microrobot that can enter cancer cells and excise a portion of the cytoplasm under the control of a rotating magnetic field [61].

### 6.3. Cell Manipulation

Manipulating cells involves adjusting their spatial placement, achieving separation from adjacent cells with distinct phenotypes, guiding them to specific target positions, or organizing them in vitro [212,213,214]. Magnetically powered miniaturized robots exhibit the capacity to manipulate a cell in three dimensions and can encompass tasks such as grabbing, conveying, categorizing, secluding, and patterning. They demonstrate outstanding agility and elevated accuracy at the microscale within intricate physiological surroundings while preserving the inherent characteristics of the cells [215]. For example, achieved by magnetically driven micromotors resembling peanuts, creating predefined patterns with cells is entailed using an organized substrate for the cell loading and subsequent delivery process [216]. Additionally, Kim et al. conducted precise manipulation of a microrobot by carrying neurons driven by magnetism, guided it to bridge a discontinuity between two neural clusters, and established connections within fractured neural networks [66].

### 6.4. Environmental Remediation

In addition to being biocompatible, recoverable, and free from toxins during magnetic manipulation, magnetic microrobots can actively navigate through waterborne pollutants to remove them via capture or degradation [217,218]. For instance, to effectively eliminate both leaked oil and microplastic pollutants simultaneously, a hollow microsubmarine with magnetic properties has been developed using natural sunflower pollen grains as a template [219]. The improved ability of pollutant adsorption in mobile microrobots is attributed to their collective behavior combined with magnetically steered agitation and surpasses that of static microrobots [158].

### 6.5. Current Challenges and Prospects

From the above discussion, it is evident that magnetic microrobots are primarily applied to the biomedical field, and they have already demonstrated the capability to navigate through various animal tissues, including blood vessels, digestive tracts, lenses, and even neural networks. Currently, most research is focused on functional validation through modeling or in vitro experiments. In the future, there will be increasing emphasis on conducting in vivo experiments to better understand the performance of magnetic microrobots within living organisms, with the ultimate goal of integrating them into clinical medical practice.

## 7. Conclusions

In conclusion, magnetic microrobots have progressed notably in terms of materials, propulsion, design, fabrication, and application. Figure 6 summarizes the current challenges and future directions in various research areas. The use of biosafe magnetic materials has spurred versatile microrobot development. Various propulsion mechanisms enable precise control and include rotating, gradient, and oscillating magnetic fields. Diverse designs provide tailored solutions, and innovative fabrication techniques showcase creativity. The versatile applications of magnetic microrobots for micro-/nano-manipulation, including drug delivery, highlight their transformative impact in healthcare, environmental management, and beyond.

Despite the successful development of various impressive magnetically actuated microrobots, most are currently limited to the proof-of-concept stage. They can only demonstrate simple functions in artificially designed simulation environments. Practical application in complex biomedical environments remains a significant challenge, indicating a substantial journey ahead.

The promising future applications of magnetic microrobots primarily include several aspects. The first one is cancer therapy. The World Health Organization has classified cancer as the second leading cause of death globally. Microrobots can be designed to locate tumor sites and deliver drugs or therapeutic agents to the tumor area, reducing damage to healthy tissues and improving treatment efficacy. Microrobots can also be designed for in vivo monitoring, diagnosis, or collection of tissue samples for pathological analysis. Additionally, microrobots can play a role in minimally invasive surgery.

The second one is thrombus clearance. Microrobots can be designed to carry drugs or thrombolytics, locate thrombus sites, release drugs, and promote thrombus dissolution. Microrobots can also assist with surgery and can even clean the inner walls of blood vessels regularly to prevent thrombus formation. The third one is the treatment of gastrointestinal diseases. Gastrointestinal (GI) diseases are diverse in type and have varied etiologies, symptoms, and treatment methods, making treatment challenging. Many diseases have similar symptoms, and some require endoscopic or other specialized examinations for diagnosis. Microrobots can be designed as part of an endoscope and can also deliver drugs to specific locations.

Fourth, to achieve the aforementioned applications, microrobots need to be precisely controlled to move through different blood vessels. Therefore, challenges such as resisting blood flow and smoothly passing through capillaries without causing blockages when using magnetic microrobots need to be overcome as part of future development.

Finally, one of the major challenges encountered in the practical application of microrobots is the issue of minimally invasive procedures, particularly concerning the removal of microrobots from biological organisms after they complete their tasks. Besides the requirement for biocompatible materials, designing removal mechanisms poses a complex challenge, especially regarding the localization and detection of microrobots inside the biological organism. It is imperative to ensure that the removal process minimally impacts biological tissues while preserving the integrity of the microrobots.

As we look forward, continued interdisciplinary collaboration and technological innovation are expected to drive further progress in this dynamic field, unlocking new possibilities and refining the capabilities of magnetic microrobots for diverse and impactful applications. 

## Figures and Tables

**Figure 1 micromachines-15-00468-f001:**
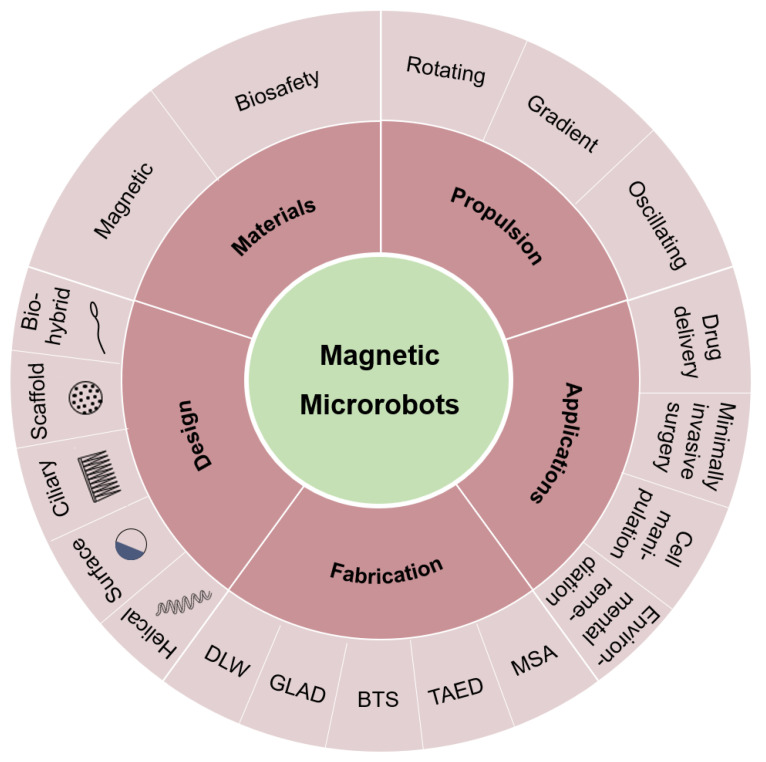
Schematic illustration of magnetic microrobots covering five aspects: materials, propulsion, design, fabrication, and applications.

**Figure 2 micromachines-15-00468-f002:**
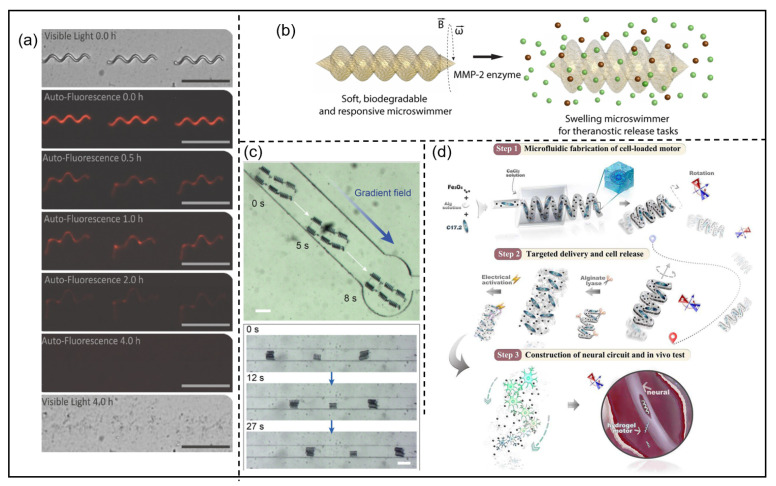
Helical microrobots: (**a**) The degradation process of superparamagnetic hydrogel swimming microrobots. © 2016 Wiley Online Library. Reprinted with permission from [83]. (**b**) A double-helical microrobot. © 2019 ACS. Reprinted with permission from [41]. (**c**) A microrobot through the depicted microfluidic tapered channel. © 2023 Wiley Online Library. Reprinted with permission from [136]. (**d**) Fabrication process of helical hydrogel micromotors. © 2024 Elsevier. Reprinted with permission from [137].

**Figure 3 micromachines-15-00468-f003:**
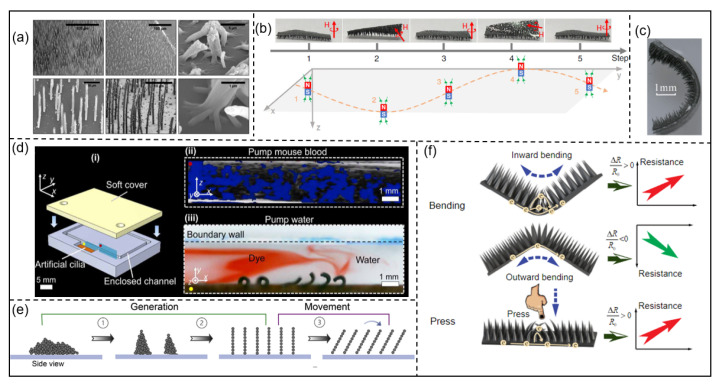
Ciliary microrobots: (**a**) Magnetic rod arrays. © 2007 ACS. Reprinted with permission from [141]. (**b**) The locomotion mode of the microrobot. © 2018 Nature. Reprinted with permission from [65]. (**c**) A bionic magnetic inchworm robot. © 2023 MDPI. Reprinted with permission from [143]. (**d**) A cilia array pumping in enclosed channels. © 2020 Science. Reprinted with permission from [144]. (**e**) Cilia’s generation and controlled movement. © 2023 Elsevier. Reprinted with permission from [145]. (**f**) Working principle of the sensor with magnetic cilia structure. © 2023 Wiley Online Library. Reprinted with permission from [146].

**Figure 4 micromachines-15-00468-f004:**
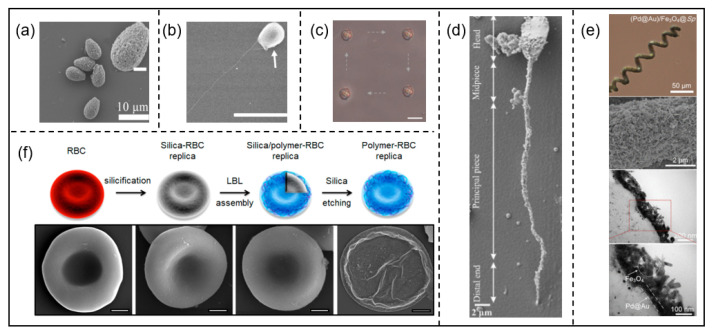
Biohybrid microrobots: (**a**) Spore-based microrobots. © 2019 Science. Reprinted with permission, from [159]. (**b**) Bacteria-based microrobots. © 2017 Nature. Reprinted with permission from [160]. (**c**) Macrophage-based microrobots. © 2017 Elsevier. Reprinted with permission from [161]. (**d**) Sperm-based microrobots. © 2020 Science. Reprinted with permission from [162]. (**e**) Spirulina-based microrobots. © 2019 ACS. Reprinted with permission from [163]. (**f**) RBC (red blood cell)-based microrobots. © 2020 ACS. Reprinted with permission from [164].

**Figure 5 micromachines-15-00468-f005:**
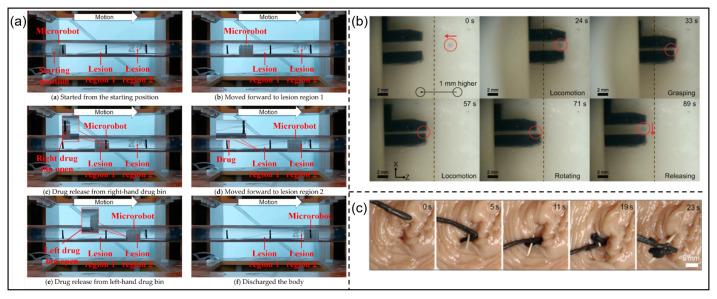
Biomedical applications of magnetic microrobots: (**a**) Targeted drug delivery. © 2021 MDPI. Reprinted with permission from [192]. (**b**) Manipulation of the zebrafish embryo. © 2023 IEEE. Reprinted with permission from [193]. (**c**) Extraction of thorn. © 2023 AAAS. Reprinted with permission from [27].

**Figure 6 micromachines-15-00468-f006:**
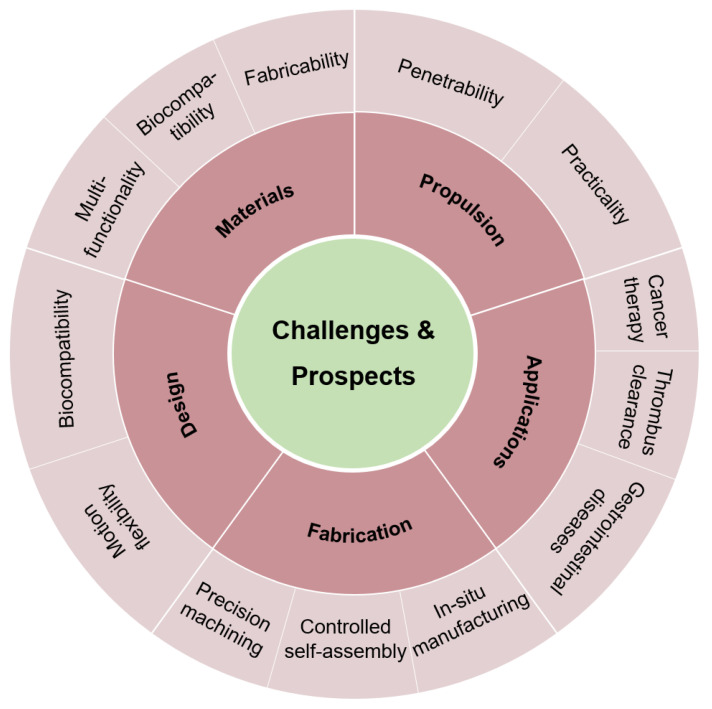
Challenges and prospects for magnetic microrobots.

**Table 1 micromachines-15-00468-t001:** Comparison of recent magnetic microrobots.

Design	Actuation Methods	Materials	Fabrication Methods	Applications	Speed (μm·$s^{-1}$)	Size (μm)	Ref.
Helical	Rotating	Fe_3_O_4_, PLA, ATBC, DCM ^1^	TAED	Drug delivery	2950	985×250	[52]
	Rotating	Photoresist	DLW	Drug delivery	128	300×10	[53]
	Rotating	NdFeB, PLA	DLW	Drug delivery	≈200–2300	3000×109	[54]
	Rotating	MNPs ^2^, PEGDA700, ethylenediamine	DLW	Drug delivery	160	100×35	[55]
	Rotating	Fe, Pt, Si$O_{2}$	GLAD	Cell manipulation	24	1.5	[56]
	Rotating	Fe, Ti, Ormocomp, PMMA, PDMS ^3^	DLW	Cell manipulation	≈800	130–170	[57]
	Rotating	Ni, Zinc-based MOF ^4^, framework-8 (ZIF-8), zeolitic imidazole	GLAD	Drug delivery	50	10	[58]
	Rotating	Fe_3_O_4_, PEGDA, PETA ^5^	DLW	Minimally invasive surgery	82	120	[59]
Surface	Rotating	Fe${\mathrm{Cl}}_{2}$·4$H_{2}$O, Fe${\mathrm{Cl}}_{3}$·6$H_{2}$O, PDMS	DLW	Drug delivery	1000–5020	–	[60]
	Rotating	Au, Ag, Ni	TAED	Minimally invasive surgery	13.19	6×0.6	[61]
	Rotating	Au, Ni, Si$O_{2}$	TAED	Drug delivery	600	3.0–7.8	[62]
	Oscillating	γ-Fe_2_O_3_, Pt, sulphonyl esters, PM ^6^	MSA	Cell manipulation	2.0±0.05	≈4.5	[63]
Ciliary	Rotating	NdFeB, Ecoflex 00-30	TAED	Drug delivery	83	4000	[64]
	Gradient	Fe, PDMS	TAED	Drug delivery	640	17×7	[65]
Scaffold	Rotation	Ni, Ti$O_{2}$, IP-S photoresist	DLW	Cell manipulation	–	95×300	[66]
	Rotating	MNP, SiCN	DLW	Cell manipulation	≈85.56	42×15	[67]
	Gradient	Ni, Ti, SU-8	DLW	Cell manipulation	≈1500	70–90	[68]
Biohybrid	Gradient	Fe_3_O_4_, bacteria, Spirulina platensis	BTS	Minimally invasive surgery	21.7–78.3	≈50	[69]
	Rotation	Fe_3_O_4_, BaTi$O_{3}$, S. platensis	BTS	Cell manipulation	333.3	≈22×0.6	[70]
	Oscillating	Fe_3_O_4_, RBCs, IGG ^7^	BTS	Drug delivery	56.5	≈2	[71]
	Rotation	Fe_3_O_4_, Pine pollen	BTS	Drug delivery	175.19	25	[72]

^1^ PLA, polylactic acid; ATBC, Acetyl-tributyl citrate; DCM, dichloromethane; ^2^ MNPs, magnetic nanoparticles; ^3^ PMMA, poly methyl methacrylate; PDMS, poly(dimethylsiloxane); ^4^ MOF, metal–organic framework; ^5^ PETA, pentaerythritol triacrylate; ^6^ PM, commercially available superparamagnetic microspheres; ^7^ RBCs, red blood cells; IGG, indocyanine green.

## Data Availability

No new data were created or analyzed in this study. Data sharing is not applicable to this article.

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
