# Peer review of "A Survey of Recent Developments in Magnetic Microrobots for Micro-/Nano-Manipulation"

_micromachines, 2024, doi:10.3390/mi15040468_

Round 1

Reviewer 1 Report

Comments and Suggestions for Authors

At the outset, it can be said that the outline of the paper makes a lot of sense. However, some works needs to be done regarding the content/flow of individual sections and subsections. Some comments for the authors:

1)      English language:

a.       Line 187-188: Cannot understand that sentence. Please rephrase.

b.       Line 228: drive actuation.

c.       Line 254: be have.

d.       Line 413-414: It seems the sentence is incomplete.

e.       Line 658: It seems the sentence is incomplete.

2)      The Abstract says that magnetic actuated microrobots are “cost-effective”. Cost-effective compared to what? It could also be argued that they are not cost-effective as it involves fancy coating or polymerization methods.

3)      Line 69-71: That sentence seems so out of place. Because magnetic microrobots can be guided by external fields, magnetic materials have undergone thorough investigation in biomedical field? Depending on what the authors would like to say, I am sure there is a better way to rephrase your thoughts.

4)      Line 80-90: Lots of repetitions.

5)      Line 103: That paragraph is talking about superparamagnetic particles. In that context, are the authors sure that at sufficiently small sizes, magnetic particles display high remanence? Is it not the opposite? Please give citations for your claims.

6)      Line 104: The entire paper mentions ferrimagnetism once here. It is perhaps not necessary for the paper at all so it may be better to remove it completely.

7)      Line 95, Line 110: Authors rightly claim that the attraction of paramagnetic particles to magnetic fields is weaker. Why is this limitation not present in superparamagnetic particles, so that they are so commonly applied in the field of biomedical research? If it is present and still superparamagnetic particles are widely used, then the justification for it has to be very clearly mentioned.

8)      Table 1: Fabrication methods are all abbreviated. At this point, the reader has no idea what any of those letters mean. Perhaps, the authors can include a legend for them as well just like they do for the materials column.

9)      Line 130-131: Repetition.

10)   Line 160-161: The work from Nelson’s group should go before the preceding sentence.

11)   Line 187: co-deposited where?

12)   Line 212: What architecture?

13)   Line 221-223: I am not sure if that sentence is true. Could the authors please back up that claim with a reference?

14)   Line 234-236: Repetition.

15)   Equation 3 is not true for all kinds of magnetic materials. Please make the necessary changes.

16)   Line 250: When the gradient vanishes, the magnetic robot need not always move parallel to the field. This is also not a correct assertion. For example, imagine a helical microrobot which has a magnetization perpendicular to its axis. Let’s say the axis is aligned with the X axis which would mean the magnetization is in the Y-Z plane. Now, if I rotate the magnetic field in the Y-Z plane, the microrobot will move in the X-direction. Therefore, not parallel to the field.

17)   Line 261: Magnetic field strength and field gradients can independently be controlled. Ideally, you can work with gradients even at 0 field strength.

18)   Line 276-278: Either discuss in full or don’t discuss at all. Discuss and decide if this is necessary to the flow of your paper.

19)   Line 308-309 and Line 631-632 are contradictory to each other.

20)   Figure 3f caption: What sensor?

21)   Line 396-399: How is a sensor relevant to this paper?

22)   Line 419: What are mineralized motors?

23)   Line 440: what dynamic phenomena? How are they important?

24)   Line 541: What biological characteristics?

25)   Section 5.3 is more related to materials than Fabrication.

26)   Section 5: There seems to be a lack of coherence in the flow of section 4, and especially section 5. For each subsection, the flow can be something along these lines: What is the method? What is the principle of fabrication? What is the resolution achievable? What materials can be used? What are the advantages and disadvantages? Two helpful and relevant examples.

Right now, these sections look like loosely connected sentences put together without a flow. The authors can of course chose to follow a completely different flow but the “story” has to be coherent.

27)   As the authors themselves know, this field is full of exciting work. Considering that, the review has too few references. There is a lot more in this field that can be highlighted.

Comments on the Quality of English Language

Please see the comments.

Reviewer 2 Report

Comments and Suggestions for Authors

1. "3 Propulsion of magnetic microrobots" is too simple and common, please provide the control conditions of different magnetic fields to help the readers to understand the control method of robots more intuitively.

2. "3.4 Current challenges and prospects" should be supplemented with the challenges and prospects of microrobots in cluster control and three-dimensional control.

3. "1 Introduction" should be supplemented with additional references.

4. The description of ferrofluidic robots in the paper is insufficient, and the following articles are suggested to supplement the description.

[1]Xie H, Sun M, Fan X, et al. Reconfigurable magnetic microrobot swarm: multimode transformation, locomotion, and manipulation[J]. Science robotics, 2019, 4(28): eaav8006.

[2]Sun M, Yang S, Jiang J, et al. Bioinspired self-assembled colloidal collectives drifting in three dimensions underwater[J]. Science Advances, 2023, 9(45): eadj4201.

[3]Fan X, Zhang Y, Wu Z, et al. Combined three dimensional locomotion and deformation of functional ferrofluidic robots[J]. Nanoscale, 2023, 15(48): 19499-19513.

Reviewer 3 Report

Comments and Suggestions for Authors

The authors presented original review  of the recent development of magnetic microrobots. This article highlights their significance in micro-/nano-manipulation, emphasizing their tiny size, untethered control, rapid response, and cost-effectiveness. The review delves into various design categories such as helical, surface, ciliary, scaffold, and biohybrid microrobots, each showcasing distinct functionalities. Fabrication techniques like direct laser writing and glancing angle deposition play a crucial role in realizing these innovative microrobots. Overall, the survey provides a comprehensive overview of the recent advancements in magnetic microrobots, paving the way for exciting developments in micro-/nano-manipulation.

Several comments can be made on the article:

1. In section 3.1. Rotating magnetic field authors write that this type of influence can be implemented using an electromagnetic coil. This is incorrect, several electromagnetic coils are needed to create a rotating magnetic field.

2. In their review, the authors missed such a promising type of world robots as active drops. It is liquid robots that are the least invasive.

3. One of the popular methods for synthesizing world robots is microfluidic synthesis. The authors do not mention it.

4. In the discussion at the end of the article, the authors missed the problem of the invasiveness of magnetic robots. It is necessary to add a section on the removal of microrobots from living organisms. This is one of the main obstacles to the introduction of microrobots.

After these minor changes, the article can be published.
